# Data Protection by Design Tool for Automated GDPR Compliance Verification Based on Semantically Modeled Informed Consent

**DOI:** 10.3390/s22072763

**Published:** 2022-04-03

**Authors:** Tek Raj Chhetri, Anelia Kurteva, Rance J. DeLong, Rainer Hilscher, Kai Korte, Anna Fensel

**Affiliations:** 1Semantic Technology Institute (STI), Department of Computer Science, University of Innsbruck, 6020 Innsbruck, Austria; anelia.kurteva@sti2.at (A.K.); rainer.hilscher@sti2.at (R.H.); or anna.fensel@wur.nl (A.F.); 2The Open Group, Reading, Berkshire RG1 1AX, UK; r.delong@opengroup.org; 3Center for Data Science, RTI International, Research Triangle Park, NC 27709, USA; 4Institut für Rechtsinformatik (IRI), Leibniz Universität Hannover, 30167 Hannover, Germany; kai.wendt@iri.uni-hannover.de; 5Wageningen Data Competence Center, Wageningen University & Research, 6708 PB Wageningen, The Netherlands; 6Consumption and Healthy Lifestyles Chair Group, Wageningen University & Research, 6706 KN Wageningen, The Netherlands

**Keywords:** GDPR, privacy, compliance verification, informed consent, standard data protection model, data sharing, data protection by design, knowledge graph, distributed systems

## Abstract

The enforcement of the GDPR in May 2018 has led to a paradigm shift in data protection. Organizations face significant challenges, such as demonstrating compliance (or auditability) and automated compliance verification due to the complex and dynamic nature of consent, as well as the scale at which compliance verification must be performed. Furthermore, the GDPR’s promotion of data protection by design and industrial interoperability requirements has created new technical challenges, as they require significant changes in the design and implementation of systems that handle personal data. We present a scalable data protection by design tool for automated compliance verification and auditability based on informed consent that is modeled with a knowledge graph. Automated compliance verification is made possible by implementing a regulation-to-code process that translates GDPR regulations into well-defined technical and organizational measures and, ultimately, software code. We demonstrate the effectiveness of the tool in the insurance and smart cities domains. We highlight ways in which our tool can be adapted to other domains.

## 1. Introduction

The enforcement of the General Data Protection Regulation (GDPR) [1] has transformed the landscape of personally identifiable information (PII) sharing (or processing). Any company (or individual) dealing with the PII of EU citizens must first obtain informed consent (Art. 6) from the data subject (i.e., an identifiable natural person according to Art. 4 (1)) if none of the other options of Art. 6 apply, regardless of their location [2]. Further, specific requirements for informed consent need to be met as well. Consent should be specific, unambiguous, freely given, and one should be able to withdraw it with the same ease as when it was given (Art. 6). However, these requirements have also posed significant challenges to companies regarding the automation of compliance verification. Several software tools for GDPR compliance have already been developed that cover some steps required for automated compliance verification (e.g., [3,4,5,6,7]). In most cases, these tools address only a specific subset of the GDPR regulations, such as audit (or auditability) [8] or only consent management. None of the surveyed tools implement a comprehensive process that translates technical and organizational measures (TOMs) into code that are required by the GDPR but are insufficiently defined.

According to the International Association of Privacy Professionals (IAPP) [9] annual privacy governance report (2020) [10], only 47% of the European companies are “fully” or “very” compliant (up from 39% in 2019). There is a need for solutions that can help companies automate GDPR compliance verification, as noncompliance with the GDPR can result in severe fines (Art. 83).

Having reviewed the existing GDPR compliance verification tools and their limitations, we have identified several issues (see Section 3) that many companies still face. A major challenge for companies, when complying with the GDPR, is to adhere to the *“principles of data protection by design”* and *“data protection by default”* (Rec. 78). Examples of such principles include the implementation of technical and organizational measures that ensure data integrity and confidentiality (Art. 32 (1) and Rec. 46) and the prevention of unauthorized disclosure or access to personal data (Art. 32 (2)), which all add another layer of complexity. A further challenge is the representation of consent in a consistent and uniform manner, which is required for interoperability and is currently lacking in existing GDPR solutions [4]. Companies also need to deal with dynamic consent (i.e., data subjects have the right to withdraw their consent at any time according to Art. 7 (3)), while remaining compliant with the regulation (or meeting compliance obligations). An example of the dynamic nature of consent is a broken consent chain due to a property ownership transfer, as described in [11], that is often encountered in the insurance domain. A system that can handle dynamic consent should be able to not only successfully detect broken consent chains but also support the re-establishment of valid consent. Finally, the scalability of processes such as automated compliance and audit is still a significant challenge, particularly in industries such as insurance and smart cities where time is critical and delays can result in additional monetary charges for businesses [8].

In light of current challenges and industrial requirements, and building on our initial idea [12], we present a scalable tool for automated GDPR compliance based on semantically modeled informed consent (part of the the smashHit [13] project). Compliance refers to the verification that a person’s data are used according to that person’s informed consent. Our work adheres to data protection by design principles by implementing a novel regulation-to-code process that translates GDPR regulations first into protection goals defined by the Standard Data Protection Model (SDM) and then into well-defined technical and organizational measures (TOMs; details in Section 4.1). Both the SDM and TOM steps of this process are executed by the legal team associated with the smashHit project. The same legal team is also supervising the implementation of TOMs into code. Our tool makes extensive use of semantic technologies, specifically a knowledge graph (KG) and an ontology (both developed in collaboration with legal experts), for representing informed consent as defined by the GDPR. Semantic technology is also regarded as one of the best practices in RegTech [14] and Ryan et al. [6]. Knowledge graphs, for example, provide a consistent consent representation in a machine-readable format, support data interoperability, and faster and easier knowledge discovery [15]. Numerous semantic models for consent exist as shown in the survey by Kurteva et al. [16] and their use for cases such as GDPR compliance verification is evident, as shown in, e.g., [17].

### 1.1. Goal

The primary objective of our work is to automate GDPR compliance verification with the help of semantics and a regulation-to-code process. To achieve this, we present a scalable, ready-to-be-deployed data protection by design tool, which utilizes a legal KG. Automated compliance verification and the handling of broken consent chains are the two main functionalities supported by our tool. The second objective is to show that GDPR regulations can be translated into code via several intermediate steps. All system requirements have been derived from our two business use cases related to data sharing (see Section A.2).

### 1.2. Contributions

With our work, we make the following contributions:We present a scalable tool for automated GDPR compliance verification based on informed consent, with a use case in smart cities and insurance (see Section A.2) that can be generalized to other domains. By “scalable”, we mean a system design (i.e., architecture) that adapts to incoming requests, such as scaling up as the number of requests increases.We present a process with a sequence of intermediate steps (regulation -> SDM—TOM -> code) that translates legal regulations into code. We show that implemented TOMs can be systematically evaluated and automatically tested (in Section 7.2).Our tool supports data interoperability through the use of semantic technology (ontology and a KG).Our tool implements a solution for broken consent chains based on two industrial use cases.

The rest of the paper is structured as follows. Section 2 provides background information on the challenges of data sharing as a result of the GDPR. Section 3 provides an overview of the related works. Section 4 summarizes the GDPR TOMs and the KG upon which our work is based on. Section 5 discusses the architecture of compliance verification tool and Section 6 provides details on its implementation. Section 7 presents the evaluation, which is followed by conclusions in Section 8.

## 2. Background

This section provides a high-level overview of the GDPR’s impact on the data sharing landscape (or data processing landscape), highlighting the importance of our work. As shown in Figure 1, we observe two distinct data sharing scenarios: (i) situation prior to GDPR implementation with no consent requirement, and (ii) situation with GDPR. Prior to the GDPR, there was no requirement for consent, and thus the data sharing (or processing) process was simple and straightforward, as the data controllers (DC) (Art. 4(7)) and data processors (DP) (Art. 4(8)) were not required to deal with consent or any regulatory laws such as the GDPR. However, with the implementation of the GDPR, the data sharing and data processing landscape has shifted and has become more complicated. This shift is illustrated in Figure 1. DP and DC are not only required to obtain consent, but they must also comply with other GDPR requirements (see Table 1) and be able to demonstrate compliance (i.e., show that they are doing only what was consented to by the data subject) to regulatory bodies upon request.

For example, it is necessary to maintain a log of data processing operations in order to demonstrate compliance by a DC (or a DP). This is to demonstrate to the appropriate authority that only consent-based processing is being carried out. Additionally, it is necessary to perform compliance checks on data sharing (or data processing) activities on a regular basis to ensure that everything is performed in accordance with consent. For example, regular checks need to verify that consents are actually still active and have not been revoked or have expired. Moreover, performing these tasks manually becomes impossible in cases such as smart cities and car insurance because of the scale at which they must be performed. As a result, automated solutions that can handle these tasks on a large scale are required.

## 3. Related Work

This section provides an overview of existing state-of-the-art solutions for GDPR compliance verification. A summary of the existing solutions and their limitations is presented in Section 3.2 and Table A1.

### 3.1. Compliance Verification

Ranise and Siswantoro [25] present a scalable approach for automated legal compliance checking based on security policies, which were translated into formal rules, in the context of the European Data Protection Directive (EU DPD) [26]. However, the work was completed before the acceptance of GDPR, which can be seen as a limitation. Further, formal rules are restrictive, and legal input is hard to implement. TOMs and semantics have not been considered.

Robol et al. [27] propose a privacy-by-design framework, which supports the development of GDPR compliant systems. The authors focus on the STS-ml extension of the Socio-Technical Security (STS) [28] method, which can be used to express and verify privacy policy rules. The implementation of a tool that enables automatic compliance based on the STS-ml is planned as future work, which can be seen as a limitation. TOMs have not been considered.

The use of semantics, specifically an ontology, for GDPR compliance is explored by Westphal et al. [29], who propose a framework that consumes and transforms privacy policies into rules. In addition to compliance based on one’s informed consent, the framework is also able to perform compliance based on relevant GDPR obligations. Evaluation details have not been presented. The limitations of the framework with regards to performance and scalability are unknown. TOMs have not been considered.

By deriving the main obligations of actors such as data protection officer (DPO), data subject (DS), and requirements for compliance that need to be met, Rhahla et al. [3] propose guidelines for implementing a GDPR compliance verification framework for Big Data systems (i.e., systems that can handle high volumes of unstructured data [30]). However, there is a lack of implementation details and information if the solution was developed with input from and validated by legal experts, which can be seen as a limitation. Further, the study did not take into account any TOMs or the use of semantic technology to solve issues such as information interoperability.

Brodin [31] present a framework for GDPR compliance to assist small and medium-sized enterprises (SMEs) in adapting to GDPR. Compliance is achieved by adopting an employee’s work style to the designed work routines, policies, and templates. The framework involves practitioners and has been empirically and theoretically evaluated. However, little to no details about the actual implementation and the used technologies are presented in [31]. Further, the framework does not provide process automation, and scalability issues are present.

Camilo [32] proposes a blockchain-based tool for GDPR consent management, which allows individuals to exercise their rights (e.g., give, revoke consent), and supports auditing of consent transactions and processing activities. Although several GDPR Articles have been followed for compliance and a proof-of-concept prototype has been implemented, specific implementation details have not been provided. The evaluation has shown that blockchain supports transparency and traceability when it comes to data processing and consent. However, blockchain’s immutability clashes with GDPR’s “rights to erasure” [33]. It is not clear if scalability and performance evaluations have been conducted and if TOMs were considered.

Arfelt et al. [8] present a tool for automated GDPR compliance monitoring, which focuses on auditing at scale. The authors use metric first-order temporal logic (MFOTL) [34] to formalize GDPR requirements and the MonPoly [35] monitoring tool to determine whether the logs confirm the given MFOTL formula and to check for violations. The work is validated by using industry data logs. However, it focuses on compliance monitoring and only implements a few of the GDPR clauses. Semantics were not utilized and TOMs were not considered.

Piras et al. [36] present an architecture for a privacy-by-design platform for GDPR compliance as part of the Data govErnance For supportiNg gDpr (DEFeND) [37] project. The platform is capable of organizing and analyzing individual’s data privacy preferences and the consent itself. Further, it is able to monitor run-time execution of functionalities and to detect and respond to security breaches. However, there is a lack of implementation details and the scalability is unknown.

Similarly to [32], Truong et al. [38] propose a blockchain-based platform for data management in compliance with GDPR, which grants and validates permissions regarding data usage by utilizing smart contracts. The performance evaluation of the implemented tool has shown that with high workload, the latency rises and bottleneck issues are present. Issues such as data interoperability have not been addressed.

Barati et al. [39], built upon their previous work in [5,40], and propose a GDPR compliance verification method in the Internet of Things (IoT) based on multiple smart contracts using blockchain (Etherium [41]). The evaluation shows that the execution time of operations such as compliance verification is highly dependent on the interest of the blockchain miners. Further, the authors acknowledge that focusing mainly on data privacy (e.g., visibility of how data is used by smart devices) and less on the security of the operations can be seen as a limitation in their work [39].

Kirrane et al. [17] present the SPECIAL-K platform for personal data processing transparency and GDPR compliance, which makes use of semantic technology, namely vocabularies. Compliance is achieved by using the HermiT [42] reasoner over semantically modeled privacy policies and event logs. In addition, Kirrane et al. propose several *“choke points”* such as increased number of users and different privacy policy complexity, which can be used for the evaluation of similar compliance frameworks. Although using semantic technology, TOMs have not been considered and latency issues are present when dealing with different complexities of privacy policies.

Bonatti et al. [43] present the SPECIAL policy language that can be used to model consent, obligations, and policies in a machine-understandable way with OWL [44] and an approaches for automated GDPR compliance verification. Compliance is performed based on ad hoc reasoning algorithms (see [45] for details). Based on the evaluation results, further process optimizations are needed. TOMs were not considered.

Mahindrakar and Joshi [46] also focus on utilizing semantics and blockchain for GDPR compliance. Legal knowledge from privacy policies is transformed into a KG, which is later integrated with blockchain to support auditing. Compliance is performed with the help of reasoning over rules related to specific data operations. Although the proposed decentralized solution eases compliance [46] and uses semantics, it also uses blockchain, which is a limitation, as discussed in [16,33]. Addressing privacy-by-design issues has been set as future work.

Barati and Rana [47] encode GDPR rules as opcodes in smart contracts in a cloud-based environment to automatically verify data operations. The evaluation of the prototype showed that as the number of operation increases, the consumed gas increases as well. The authors found a correlation between the fee paid by a data subject and the rate of violation detection—the violation detection increases when the number of operations decreases [47]. A current limitation of the study is that it has not yet investigated how the “right to be forgotten” can be implemented, having in mind the use of blockchain.

Ryan et al. [6] present a set of requirements for GDPR compliance based on RegTech [14] and a prototype implementation of a GDPR compliance verification tool that can assist DPOs in maintaining GDPR accountability. The authors utilize semantics (the Data Privacy Vocabulary (DPV) [48] and PROV-O [49] ontologies). The evaluation showed that the tool helps achieve 100% compliance for accuracy, retention, and security, only 50% compliance for data breaches, and 40% regarding data subject rights. However, TOMs were not considered and the scope of the work is limited to providing information needed for compliance verification to DPOs.

Semantics and blockchain are also used by Merlec et al. [7], who present a dynamic consent management system for personal data usage in compliance with GDPR. The system supports functionalities such as data provisioning, user management, and authentication. Some of the known limitations, as mentioned by the authors in [7], include the complexity of the presented solution, immutability of blockchain, and the automation of security and privacy policies verification and GDPR compliance checking, which is planned as future work.

Hamdani et al. [50] combine machine learning (ML) with rule-based reasoning and propose a framework for automated GDPR compliance checking. Compliance is based on Article 13 and 14 and uses the OPP-115 [51] taxonomy to capture 10 rules from these articles. The combination of rules and ML has shown to accurately predict both coarse-grained and fine-grained data processing. However, the framework is limited with regards to the types of documents it covers. The use of multiple compliance documents, such as data protection impact assessments (DPIAs) [52], is set as future work, and TOMs were not considered.

Similarly to our work, Daoudagh et al. [53] also focus on data sharing in smart cities and propose a privacy-by-design platform for compliance. The implemented solution preserves individuals’ privacy by utilizing various authentication mechanisms, such as zero-knowledge proofs, it can support data management and traceability, and it is compliant with GDPR principles such as data minimization and purpose limitation. Although the system has been implemented and utilizes the DPV [48] ontology, details about how exactly compliance verification is performed have not been provided, and only two TOMs have been considered.

Finally, Tokas et al. [54] propose a policy language that models different GDPR principles, purposes, access rights, and a static rule-based approach for compliance based on specific privacy policies. Although the work is based on several GDPR articles, including data protection by design, improvements are needed regarding the language’s expressivity (e.g., to model DC (or DP), data retention periods). Further, the proposed static mechanism does not fully automate GDPR compliance, and TOMs have not been considered.

### 3.2. Summary

This section presents a comparison of our solution to the reviewed state-of-the-art projects. Table A1 (see Section A.3) highlights several characteristics that we have derived from our two use cases and the need for a scalable GDPR compliance solution that supports data interoperability and compliance verification automation. Given the volume of processing required for use cases such as smart cities, we considered scalability as a critical factor. When comparing the solutions, we also considered the implementation of GDPR-required TOMs. Details on TOMs are presented in Section 4.1 and Section 7.2. The industrial use case provides information on whether the work was conducted based on real industrial use cases. Finally, we considered the implementation level of the solutions as it helps distinguish between ready-to-be-deployed and proof-of-concept solutions. In Table A1, a check mark indicates that a feature has been addressed, while a cross mark in red color indicates that a feature is either not implemented, there is a limitation, or it is unknown (i.e., no information was provided in the study).

The details on the implementation notation used in Table A1 are presented below.

**P** indicates that GDPR compliance has only been proposed or guidelines provided without actually implementing a proof of concept or implementation.**PoC** indicates the completion of a proof of concept (or experimental implementation that is not at the prototype level).**PT** indicates that a prototype implementation has been completed.**FM** indicates ready-to-deploy work that can be integrated via Representational State Transfer (REST) Application Programming Interface (API). With ready-to-deploy code, we refer to an implementation that can be deployed and integrated with minimal configuration.

As shown in Table A1, the solutions which utilize semantics and allow consent creation/revocation do not fully automate GDPR compliance and do not consider TOMs. The studies which automate compliance have experimental PoC implementation; thus cannot be directly deployed and used. Although all these solutions have undergone performance evaluation, a scalability evaluation was performed only for [46,50]. Apart from [25], none of the studies discussed the ease of adapting their respective approaches to other regulatory frameworks. Most importantly, a review of related works reveals that the majority of current studies lack consent creation and revocation features. In comparison to all of the studies in Table A1, our solution is not only capable of performing automated GDPR compliance verification but it also implements a robust regulation-to-code process that translates GDPR regulations into well-defined TOMs via the SDM. It also supports interoperability by using a KG. Further, it has undergone extensive performance and scalability evaluation based on real-world industrial use cases and has a PT implementation, which allows it to be deployed and used straight away. The following sections present the details about its design, implementation, and evaluation.

## 4. KG Overview and Legal Background

This section presents details about the two main tasks that allow us to implement a scalable automated GDPR compliance verification tool based on semantically modeled informed consent. A tool that implements a set of legal regulations (such as the GDPR) requires a way of translating these regulations into a unified machine-readable format. Modeling and verifying consent across a variety of domains (e.g., our two uses cases discussed in Section 1 and Section A.2) can be achieved with the help of a semantic data model (e.g., the KGs). Our approach to address the regulations-to-code task adds a software implementation step to the previously established methodology that translates vaguely-defined GDPR TOM requirements to so-called protection goals defined by SDM [55], and finally, well-defined TOMs are important as they enable data protection by design, that can be implemented in code. All legal entities that filter into the compliance verification process (including protection goals) need to be semantically represented and readily accessible to other integrating components. A particularly flexible and extendable way of doing so is a legal KG.

### 4.1. GDPR and Relevant TOMs

Our GDPR compliance verification tool follows the *“data protection by design and by default”* principle to ensure GDPR compliance (Table 1). A key instrument here is the implementation of *“appropriate technical and organizational measures which are designed to implement data protection principles […] in an effective manner and to integrate the necessary safeguards into the processing in order to […] protect the rights of data subjects”* (Art. 25 (1)). This includes the adoption of internal policies (Rec. 78). It is the responsibility of the DC (or DP) to implement TOMs to ensure and to be able to demonstrate that processing is performed in accordance with the GDPR (Art. 4 (7), Art. 24). The implementation of TOMs also mitigates the risk to the rights and freedoms of natural persons posed by the processing of their personal data (Art. 32 (1)).

Legal requirements are not easily translated into technical implementations. The SDM [55] was developed to serve this exact translation function. The SDM translates legal requirements into corresponding protection goals that can be directly implemented as TOMs. GDPR requirements and their mapped protection goals are summarized in [55] (Table in Section C2, p. 28). The SDM:Systematizes data protection requirements in form of protection goals.Systematically derives generic measures from the protection goals, supplemented by a catalog of reference measures.Systematizes the identification of risks in order to determine protection requirements of the data subjects resulting from the processing.Offers a procedure model for modeling, implementation, and continuous control and testing of processing activities.

Table 1 lists each SDM protection goal and its relation to the respective GDPR principle(s). Further, Table 1 associates protection goals with specific TOMs and their current implementation in the automated GDPR compliance verification tool.

### 4.2. Legal KG

The main data source for performing compliance verification is a KG, which is based on the smashHitCore [56] ontology and is stored in the GraphDB graph database (see Figure 2). The KG models informed consent as defined by GDPR Art. 7 and Rec. 32. Concepts such as contracts, sensor data, data processing, and the involved entities are represented as well. When it comes to the consent itself, the KG represents its (1) state (e.g., granted/not granted, revoked, withdrawn), (2) purpose, (3) duration, (4) the requested data and its type (e.g., sensor data), (5) types of data processing (e.g., analysis, retrieval, adaptation, collection) associated with consent, (6) entities related to consent (e.g., DC (or DP), data subject, etc.), and (7) the time and date of a consent state change.

Consent is collected via consent forms, which have been evaluated against GDPR’s requirements for informed consent (Art.7, 12, 13, and Rec. 32) by the legal experts (i.e., manual evaluation) participating in the smashHit project. With the help of APIs, the data from the consent forms are sent to predefined Simple Protocol and Resource Description Framework Query Language (SPARQL) [57] queries, which annotate and create unique consent instances in the KG. The KG can be accessed directly via the SPARQL API [58] provided by GraphDB. The advantages of using semantic models such as KGs for consent are also discussed in [59,60].

## 5. System Architecture

In this section, we present the details of our architecture for an automated GDPR compliance verification tool, which addresses current data sharing and processing challenges (see Section 3). The main goal of our work is to simplify and automate GDPR compliance verification with the help of semantics and a regulation-to-code process, as shown in Figure 1 (see Section 2).

Figure 2 presents the architecture of the automated GDPR compliance verification tool, which follows a microservices architecture pattern. A microservices architecture pattern is one in which all modules are cohesive, independent processes that interact through messages [61]. This specific architectural pattern provides a solution to problems such as technology lock-in for developers and dependency problems that monolithic patterns suffer from [61,62]. Moreover, the microservices pattern supports scalability (one of the challenges and industrial requirements in Section 1) by allowing for scaling of individual small components [63,64]. Additionally, the flexibility and maintainability of microservices architectures are reasons to consider them.

### 5.1. Service Layer

The service layer implements core functionality required for automated compliance verification and related supporting operations such as consent creation. It is divided into two parts: (i) the *API layer* and (ii) the *core*. The *API layer* and *core* components are discussed in detail in the subsections below.

#### 5.1.1. API Layer

The API layer serves as the primary entry point for the tool and exposes REST endpoints. The API layer provides access to the compliance verification tool’s functionalities via REST endpoints. In addition to exposing the tool’s access, one of the major features of the API layer is the ability to enable role-based access, which prevents unauthorized access and is one of the GDPR TOMs (see Section 4.1). Algorithm 1 (more details in implementation Section 6.2.1) shows the steps for enabling role-based access. The organizational code in Algorithm 1 is a unique identifier assigned manually (i.e., by an administrator) and is provided to other parties accessing the compliance verification tool.
**Algorithm 1:** Role-based API endpoint access. 
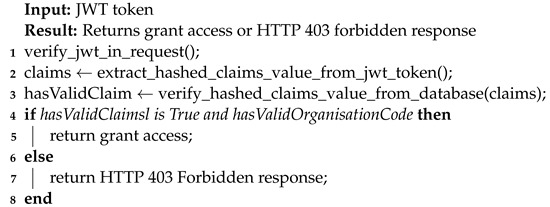


#### 5.1.2. Data Processing

The data processing module is responsible for data management in order to support required operations, such as compliance checking, consent creation, and auditing. The data processing module is further subdivided into two components, the query processor and storage, according to the type of data processing operations it supports. The query processor module contains the SPARQL queries required to deal with consent (i.e., consent data in KG), while the storage module handles the query processor’s execution.

#### 5.1.3. Shared Service

The shared services include modules that assist other modules in their operations, such as compliance, consent, and security. This shared service module’s purpose is to increase reusability by abstracting away common functionality from other modules. As with data processing, the shared services module is composed of two subcomponents: helper and CRON. The CRON is a unique shared service shared by the scheduler and compliance that initiates an automated compliance check in response to the scheduler’s trigger. The helper, however, is shared across multiple modules, such as consent and security.

#### 5.1.4. Security

The security module incorporates safeguards such as encryption to ensure confidentiality, which is also one of the TOMs (see Section 4.1). Furthermore, the security module supports the GDPR’s *“data protection by design”* and *“data protection by default”* principles. To support *“data protection by design”*, the security module includes two critical features: deterministic layered encryption and a decryption scheme, the architecture of which are illustrated in Figure 3 and Figure 4, respectively. Both the encryption and decryption schemes employ a hybrid algorithm (implementation details in Section 6.2.4). This is motivated by the established benefits of hybrid (i.e., using symmetric and asymmetric encryption techniques) approaches, as demonstrated by Zou et al. [65]. Furthermore, deterministic encryption enables secure querying of data over the database, and layered encryption strengthens the security, which can be increased or decreased as required.

#### 5.1.5. Consent

The consent module assists in ensuring that only necessary and required consent information is used, thereby minimizing data usage, which is one of the GDPR’s TOMs (see Section 4.1). Figure 5 shows the consent JSON schema used by the consent module and its representation in the legal KG. In particular, the consent module carried out this task of ensuring that only required information is used by performing JSON schema (see Figure 5a) validation. The JSON schema was created according to legal KG and, therefore, validation of the JSON schema ensures that consent (i.e., input consent in JSON format) is in accordance with legal KG. The steps for performing the consent validation are depicted in Algorithm 2 (implementation details in Section 6.2.5). The consent module also performs the task of transforming consent information in JSON format into the KG. The reason for this approach (i.e., not taking KG directly as opposed to JSON) is that not all businesses, particularly small and medium-sized enterprises (SMEs), have expertise in semantic technology. Thus, using the JSON format for interaction reduces the complexity and overhead associated with dealing with semantics, while allowing SMEs to take advantage of its benefits.
**Algorithm 2:** Consent validation for consent creation. 
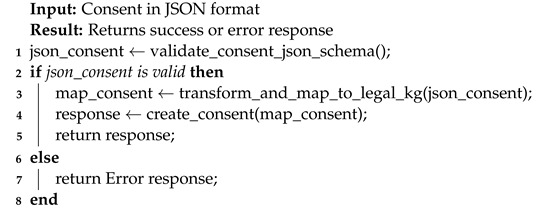


The consent validation (i.e., validate_consent_json_schema), for example, validates that all the required fields, such as purpose information, are present. Further, the consent validation validates that the consent input has the valid required data types. For instance, the consent granted and the expiration date must be in a valid date and time format and *DataProcessing* should be a list of string. Similarly, the transformation to legal KG step (i.e., transform_and_map_to_legal_kg) transforms the JSON input to the KG representation by adding appropriate details. For example, converting date and time format JSON input (e.g., consent granted time) to SPARQL date format [66] and adding an appropriate relationship, such as hasPurpose (see Figure 5b) for the purpose of consent input. Moreover, the transformation step also performs additional validation, such as checking if the consent granted time is in a specific date format (e.g., universal time format).

The *Agent* in the consent JSON schema represents the DC (or DP)’s information. The *Resource* contains information about the actual data collected from the data subject that the data processor or controller desires for data processing purposes, such as GPS data. The *DataProcessing* signifies the purpose for which the data will be used, for example, marketing.

#### 5.1.6. Auditing

One of the core principles of the GDPR is transparency (see Section 4.1), which requires that information about the processing of personal data be readily available. Furthermore, transparency aids accountability, which is defined as demonstrating compliance with GDPR obligations in a measurable manner [67]. The auditing module promotes accountability by enabling access to data processing details such as whether valid consent was obtained, whether data processing was carried out in accordance with the consent, and whether no data processing occurred in the event of consent expiration or revocation. Additionally, this information assists a DC (or DP) in demonstrating compliance, as it is the information required to be presented to authorities (implementation and auditing examples in Section 6.2.6). The auditing module, which makes the personal data processing information available to both the DC (or DP) and the data subject, is critical to transparency and is the module’s major functionality.

#### 5.1.7. Compliance

The compliance module is the heart of the system; it conducts compliance checks to ensure that DC (or DP)) have obtained informed consent from data subjects for both accessing the data and the processing applied to it (i.e., no consent is violated). A consent violation occurs when the DC (or DP) acts in a manner inconsistent with the consent. For instance, if a user (or data subject) has provided informed consent for the DC (or DP) to collect and use mobile Global Positioning System (GPS) data for research purposes, the DC (or DP) should collect and use GPS data exclusively for research purposes. The use of data for purposes other than research and the collection of data other than GPS would constitute a violation of consent, a punishable offense under the GDPR. Furthermore, in the case of automated compliance check (i.e., the compliance check instantiated by the scheduler based on scheduled time), the detection of a consent violation triggers a notification alert (details in implementation Section 6.2.7), thereby allowing DC (or DP) to take appropriate actions. While in the case of manual compliance checks, such as the one instantiated by the DC (or DP) for a particular data subject or consent, no alert is triggered.

As illustrated by Equation (Equation 1), compliance is the conjunction of the core compliance decision and the security and privacy compliance decision. The compliance module is responsible for the core compliance decisions, and the security and privacy module for security and privacy compliance decisions (details in Section 5.2). The core compliance decision is the combination of consent validity, data processing purpose, data processing rights, and data processing operations. Simultaneously, security and privacy compliance decisions are made in conjunction with the data subject’s privacy preferences and the DC (or DP) privacy policy.
(1)compliance=(core_compliance_decision∧security_and_privacy_compliance_decision)

Let ds represent the data subject and dc_dp the DC (or DP). Similarly, let $consent_{ds}=\left\{C_{ds},P_{ds},R_{ds},D_{ds}\right\}$ represent the consent information from data subject and $processing_{dc\_dp}=\left\{P_{dc\_dp},R_{dc\_dp},D_{dc\_dp}\right\}$ the data processing information obtained from the DC (or DP), where *P* denote the purpose, *R* the resources from which the data is collected (e.g., GPS), and *D* the information about the actual data processing to be carried out (e.g., data analysis). The *C*, on the other hand, contains information about the expiration of the granted consent. The core compliance decision then can be represented as shown by Equation (Equation 2). The consent validity check verifies that the consent is still valid and has not expired, while the purpose validity, processing rights validity, and data processing validity checks compare the consent to information obtained from the data processor (or controller).
(2)$$\begin{array}{cc} core\_compliance\_decision & =hasValidConsent\left(C_{ds}\right)\wedge hasValidPurpose\left(P_{ds},P_{dc\_dp}\right)\\ \; & \wedge hasValidProcessingRights(R_{ds},R_{dc\_dp})\\ \; & \wedge isDoingValidDataProcessing(D_{ds},D_{dc\_dp}) \end{array}$$

Compliant behavior necessitates satisfying Equation (Equation 1), which begins with evaluating the core compliance decision (i.e., Equation (Equation 2)). The process of determining compliance is immediately halted if any of the conditions are determined to be false, as compliance requires satisfying all of the conditions. The notification alert is then passed, providing the detailed information, such as where the compliance check failed or everything is fine in the case of success.

Consider the following consent information: (i) purpose = safety research on driving, (ii) granted time = 2018-08-22:18T08:22:01.3474Z, (iii) resources = GPS, and (iv) data processing activity = data analysis. Consider the following information as information about processing obtained from a DC (or DP): (i) resources = GPS, VSS (vehicle speed sensor), (ii) purpose = safety research on driving, and (iii) data processing activity = data analysis.

**Example** **1.**
*Assume for the moment that the scheduler is set to run the compliance check daily at time 21:01.3474Z. The current time is 2019-05-22:18T08:21:01.3474Z and the expiry time for the consent is $C_{ds}$ = 2019-05-22:18T08:20:01.3474Z. When the compliance check operation is performed (i.e., when Equation (Equation 1) is evaluated), the compliance module determines that the consent has expired (i.e., at $hasValidConsent(C_{ds})$) and that no data processing should be permitted. The compliance module begins by updating the consent status to invalid and notifying the DC (or DP).*


**Example** **2.**
*In this example, assume that consent is valid (i.e., it has not expired or has not been revoked). The DS has granted consent for only GPS data (i.e., $R_{ds}$ = GPS) to be collected (i.e., the DC (or DP) has only GPS data processing rights). The DC (or DP) does, however, collect data from both GPS and VSS (i.e., $R_{dc\_dp}=\left\{GPS,VSS\right\}$). The compliance status will then be invalid during the compliance check (i.e., at $hasValidProcessingRights(R_{ds},R_{dc\_dp})$). A similar situation can also exist if all core compliance decisions were to pass but security and privacy decisions were to fail.*


In the case of the broken consent chain, a similar compliance check step is performed, thereby invalidating the existing consent. For example, DS A, who has given consent to share car data, sells the car (which is the source of the data) to a new DS B. Here, the existing consent chain should be invalidated as it is no longer valid. Upon receiving this update via an API, the compliance tool invalidates the existing consent. However, in the case of revocation, the DS must initiate the compliance request by providing specific consent information such as a consent ID. Our tool will invalidate the consent after performing the necessary checks.

### 5.2. Security and Privacy

Security typically concerns itself with the objectives of *confidentiality*, *integrity*, and *availability* of *data*. Privacy adds to these objectives a concern for the right of a person (*DS*) to authorize and restrict the use of data that are *about*, or describe, that person to some extent (*personal data*). The common aspects of security and privacy induce us to establish a common framework within which to conceptualize and implement their related objectives. When implementing security and privacy, we exploit the distinction between *mechanism* and *policy*. Security and privacy policies are expressions of specific rules governing permissions and restrictions to the access and use of data. Security and privacy mechanisms implement data controls that may be applied in practice to enforce a policy. Ideally, mechanisms are built to not embody a particular policy but rather a policy framework, and thus to be able to interpret and enforce any policy expressed within that framework. The benefit of this approach is that it provides the flexibility to make a broad range of modifications to policy without the need to revisit the implementation of the mechanism.

#### 5.2.1. Policy Tools

The policy tools provide a policy specification language for expressing security and privacy policies for representing *privacy preference* and *privacy policy* within a framework based on the Next Generation Access Control (NGAC) standard [68]. A preference on the part of a data subject represents rules that constrain the handling of its data that the data subject has expressed through a process of informed consent. A privacy policy on the part of a data processor represents claims concerning what processing will be performed on the data subject’s personal information, the purpose for which it is performed, and commitments about the handling of personal data by the data processor and sharing with third parties.

#### 5.2.2. Policy Enforcement Points (PEP)

When a data processor seeks to perform a data processing operation on an item of personal data associated with a data subject, the processor calls a policy enforcement point to access the data. Although the PEP operates with the privilege to access the data of the data subject, it is *trusted* to make that data available to its caller only if a query to the policy decision point, citing the processor, the processing to be performed, and the purpose of the processing, returns with a “grant” response. There may be multiple PEPs in a system, or a PEP may even be within a remote data processing system.

#### 5.2.3. Policy Decision Points (PDP)

Policies, including data processor privacy policies and data subject privacy preferences, are stored within the PDP mechanism for use to compute response to queries from PEPs. A query asks whether a specific operation on a particular data subject’s personal data by a particular data processor is permissible under the policy, and if so, what the obligations are that the data processor must fulfill after it uses the data.

#### 5.2.4. Compliance-Related Policy Decisions

The PDP provides several policy query interfaces based on privacy preferences and privacy policies. The queries carry out computations over policy representations that are constructed from information gathered from the smashHitCore ontology [56] and from the KGs. The first of these queries may come from the component tasked with obtaining consent. It utilizes the combination of a data subject’s current privacy preference (as resulting from a consent) and a DC (or DP)’s privacy policy. The returned information may be used by the consent-gathering user interface to confirm that the privacy policy of a data processor complies with the data subject’s current preference, or to highlight the areas of noncompliance. This information may be provided to the data subject for consideration in modifying the subject’s consent/preference or in denying consent.

The second of the queries may come from any component tasked with responding to a request from a data processor for permission to perform a particular data processing operation on the personal data of a particular data subject. The computation of this query uses the processor’s prior declared privacy policy and the data subject’s current preference (corresponding to a valid consent) to determine whether the data processing operation, and its purpose, conform to the parameters granted in the data subject’s consent. The returned value is a “grant” or “deny”. In the case of “grant”, the return includes any obligations that the processor must fulfill following the data access/use. A “grant” response may also include a *warrant*, which is a unique token that may be presented to the PEP that is enforcing access to the requested data item.

### 5.3. Scheduler

The scheduler supports time-based event scheduling. The scheduler is pivotal in our work because it enables the automated compliance check to be performed on a regular basis, based on the scheduled time using the shared service CRON.

### 5.4. Remote Storage

The remote storage facilitates storing of the consent and the compliance verification tool decision for purposes such as auditing. The graph database and the NoSQL database [69] are the two types of databases that make up the remote storage. The graph database is because of the involvement of semantic technology (i.e., KG). Furthermore, the reason for using an NoSQL database is because of the scalability it offers and the schema-less data storage feature that is suitable for storing the information logs.

### 5.5. Serverless Layer

The serverless layer provides the serverless function for logging and querying compliance verification tool decisions, such as consent revocation, that are necessary for auditing. The reason for the use of serverless is the features serverless offers, such as scalability and ease of deployment [70].

## 6. Implementation

This section provides detailed information on the implementation of the automated GDPR compliance verification tool. Section 6.1 details the libraries used, and Section 6.2 provides details on the implementation of individual components of the tool.

### 6.1. Experimental Setup

Table 2 summarizes the libraries that were used in our implementation. The selection of these libraries was made based on the requirements of our tool. For example, GraphDB was selected for its ontological reasoning capabilities required for a KG and SQLite for providing a lightweight relational database setup. Similarly, MongoDB was chosen due to its support for horizontal scalability and its suitability for storing logs. In our implementation, MongoDB Atlas [71] was used to run three replicas hosted on Amazon Web Services (AWS) [72].

The service layer is deployed using a Docker container in a system with 16 GB (gigabyte) random-access memory (RAM), a 2.3 gigahertz (GHz) Quad-Core Intel Core i7 processor, and 500 GB storage. However, for scalability testing, the service layer was deployed in a Kubernetes [73] cluster with a total of 12 CPU (central processing unit) cores, 24 GB RAM, and 480 GB storage. The cluster was divided into three nodes, each with four CPU cores, 8 GB of RAM, and 160 GB of storage. For the Kubernetes cluster, we used Linode [74], which provides infrastructure as a service. The security and privacy component is deployed on a virtual machine with two virtual CPUs, 4 GB of RAM, and 40 GB storage. Similar to the service layer, the OpenFaaS layer is deployed on a virtual machine using a Docker container. The virtual machine has four virtual CPUs, 8 GB of RAM, and a storage capacity of 12 GB. GraphDB is installed on a server powered by an Intel Core i9-9900K processor, 64 GB of RAM, and a 1 TB NVMe storage. All of the deployment setups use Linux [75] with variant distributions such as Ubuntu [76] and Debian [77].
sensors-22-02763-t002_Table 2Table 2Information about the software (or libraries) that were used in the implementation.Software (or Libraries)VersionPython [78]3.8SWI-Prolog [79]7.6.4Flask [80]1.1.2Flask-RESTful [81]0.3.8Flask-SQLAlchemy [82]2.5.1Requests [83]2.25.1Flask Apispec [84]0.11.0Pycryptodome [85]3.10.1Flask-JWT-Extended [86]4.2.1FuzzyWuzzy [87]0.18.0NLTK [88]3.6.2Spacy [89]3.0.6SPARQLWrapper [90]1.8.5PyMongo [91]3.11.4Docker ([92] Community Edition)20.XSQLite [93]2.6GraphDB free edition9.4.1MongoDB [22]5.0.6

### 6.2. Automated GDPR Compliance Verification Tool Implementation

This section provides detailed information about the implementation of the components that make up our tool, such as the API and compliance layers.

#### 6.2.1. API Layer

The API layer implements the REST endpoints. Furthermore, the API layer provides a custom JWT implementation (see Algorithm 1) to enable role-based access as demanded by GDPR’s integrity and confidentiality principle (Art. 5(1)(f)) by extending Flask-JWT-Extended. JWT is an Request for Comments (RDF) 7519 [94] open industry-standard method for securely representing claims between two parties. As demonstrated in Algorithm 1, the hashed claim value is extracted first from the JWT token after verifying JWT in the request (i.e., a valid JWT token exists in the request header). The extracted claim value is then compared to the database containing the user’s information. Additionally, a check is made against the organization’s unique identifier. After passing both checks on hashed claims and organizational identifiers, the valid token is returned, enabling grant access. If the request fails, an HTTP 403 Forbidden response is returned, effectively blocking access to the compliance verification tool (or REST endpoints). This customized implementation of JWT to enable role-based access is designed with a focus on user-friendliness and thus follows a similar approach to enabling JWT via decorators. As a result, enabling role-based access is as simple as including the appropriate decorator, which is accomplished by using the syntax *@decorator_name()*. Furthermore, our API layer implementation uses standard REST practices such as the OpenAPI [95] Specification (OAS) version 2.0 and Swagger to describe the REST endpoints (see Figure 6).

Figure 7 summarizes our role-based endpoint access implementation. As shown in Figure 7b, valid JWT credentials must be transmitted in order to obtain a JWT token; thus, a registration (with user credentials such as username and password and an unique organizational identifier) is needed. Figure 7a shows the sequence diagram for JWT user (human or software agent) registration. The organization’s identifier is assigned manually (i.e., by an administrator) and is provided to other parties accessing the compliance verification tool. Using the organizational identifier, a unique hash token (or role or role token) generated based on timestamp is assigned. This hash token, together with the organizational identifier, is used to verify the role and access authorization to the appropriate REST endpoint(s).

#### 6.2.2. Data Processing

The implementation of the data processing module consists of predefined SPARQL queries with the necessary consent information to be filled in during run-time. The queries are organized according to the various operations supported by the tool, such as auditing and compliance checking. Figure 8 illustrates a snapshot of the SPARQL query used to obtain consent ID information as part of the data processing module’s query processor component.

#### 6.2.3. Shared Service

The shared service module implements functions, such as function_map and list_to_query (see Figure 9), that are used by other modules, such as data processing, thereby improving code reusability and abstracting complexity. The function_map performs the mapping to the actual function, while the list_to_query function converts the array of JSON inputs into the SPARQL query format for supporting consent creation activities by the consent module.

#### 6.2.4. Security

The security module implements encryption and decryption following the encryption and decryption architectures presented in Section 5.1.4 by using Pycryptodome. Asymmetric encryption, RSA [23], is used as Algorithm 1 (see Figure 3) and a symmetric encryption, AES [24], as Algorithm 2 (see Figure 4).The reason for the selection of RSA is because of its proven capability and security robustness over the last nearly 30 years. The reason for considering AES, an encryption technique standardized by National Institute of Standards and Technology (NIST) [96], is that AES is considered the de facto standard for symmetric encryption and it is fast and secure [97]. When asymmetric encryption is used, storing encrypted data in a database presents a challenge when it comes to querying the data. The encryption must be deterministic, which is why symmetric encryption was chosen to encrypt the actual data as in Algorithm 2. Our AES implementation allows for searchable encryption while preserving the required confidentiality.

The RSA implementation uses the PKCS #1 OEAP [98] padding scheme, which is defined by RFC 8017 [99]. RSA is used in our implementation to encrypt and decrypt the keys of the symmetric encryption algorithm. The RSA key is generated dynamically at run-time, the first time the application is deployed, if the key does not already exist. A key with a length of 1024 bits is generated and exported. The exported key is password protected using PKCS#8, an RFC 5208 [100] standard for storing and transferring keys. To protect the generated RSA keys, we used the PKCS#8 protection scheme *scryptAndAES128-CBC* in our implementation. Similarly, the AES (or Algorithm 2) implementation, as shown in Figure 3 and Figure 4, is multilayered, three in our case. For example, when our layered AES implementation generates the ciphertext ***C***, the plain text ***P*** is passed through multiple encryptions (E) as E(E(E(P,k1,iv1),k2,iv2),k3,iv3) using unique secret keys K={k1,k2,k3} and IV keys IV={iv1,iv2,iv3} at each step. This multilayered AES encryption enhances security and makes it more difficult for attackers to decrypt the encrypted ciphertext. In the case of decryption (D), similar steps as in encryption are performed, but in the reverse order, as D(D(D(C,k3,iv3),k2,iv2),k1,iv1), to obtain the plain text P. By changing the layers in the security module, the AES layers can be adjusted (increased or decreased) as needed. A 32-bit secret key and a 16-bit IV key are used in our AES implementation. The implementation also employs Bellare et al. [101] encrypt-then-authenticate-then-translate (EAX) mode, which enables both authentication and privacy of the encrypted message. Moreover, when the application is deployed, all of the keys for each AES layer are generated dynamically during runtime, just like RSA if the key does not exist.

#### 6.2.5. Consent

The implementation of consent module comprises the realization of Algorithm 2, which performs the consent validation, taking consent in a consent JSON format following consent JSON schema (see Figure 5a) and transforming it into KG for consent creation. The marshmallow [102] is used to validate consent (i.e., the consent JSON schema validation). Marshmallow is a framework-agnostic object serialization library for converting complex data types, such as objects, to and from native Python data types [102].

Figure 5b shows the KG representation (or creation) of informed consent, while Figure 10 depicts the response that is sent following a successful creation of a consent instance. Additionally, as illustrated in Figure 5b, the consent information is encrypted. This is to prevent gaining insights about the data subject.

#### 6.2.6. Auditing

The auditing module implements two categories of auditing, namely, partial auditing and full auditing. Both the partial and full audits can be conducted with consent or by the data provider using a unique consent (or data provider) identifier. The partial auditing provides only high-level information, such as consent decision and its status (e.g., revoked). Full auditing provides complete information including the consent itself, consent status, and compliance check decisions. Figure 11 and Figure 12 show the partial auditing and full auditing response, respectively. Moreover, auditing based on a single consent only provides information about that individual consent, whereas auditing based on the data provider provides information on all consents provided by that data provider.

#### 6.2.7. Compliance

The compliance module implements three different levels of compliance checking, namely, consent-based, data provider-based, and automated compliance checking by executing Equation (Equation 1). Similar to consent-based auditing, consent-based compliance checking performs a compliance check of the single consent. In contrast, data provider-based compliance checking performs a compliance check on all the consents associated with the particular data provider whose compliance is being checked. In the case of automated compliance checking, however, the compliance check is performed for all active consents. The sample responses to compliance checks based on data provider, individual consent, and automated compliance checks are shown in Figure 13a,b and Figure 14. Additionally, as discussed in Section 5.1.7, when an automated compliance check is performed, a notification alert is created. This requires configuring an external notification URL to receive notifications about the result of the compliance check. The external notification URL is a URL that the DC (or DP) must provide in order for the alert to be received. No notification is sent in the case of consent-based and data provider-based compliance checks. This is because consent-based and data provider-based compliance checks are initiated manually via REST endpoints and the result of the compliance check is returned as a response to the same REST endpoint that initiated the check.

The implementation employs a fuzzy string matching technique based on the Levenshtein distance (i.e., edit distance) and computes the token set ratio, which is used to determine the match’s similarity. FuzzyWuzzy [87], a fuzzy string matching Python library, was used to implement a fuzzy string matching technique. Additionally, natural language processing techniques such as tokenization and stemming were used to address grammar and spelling errors.

#### 6.2.8. Security and Privacy

The security and privacy implementation extends the NGAC languages and tools for privacy policy specification and enforcement, by the inclusion of *purposes* for processing of personal data, *retention* periods for data, fine-grained data processing operations, and other concepts common in extant privacy policy frameworks. In addition to the extension to the NGAC declarative policy language, the implementation of the security and privacy modules also includes an experimental implementation of the SecPAL for Privacy (S4P) language [103] that provides more explicit representation of delegation and distinction of roles in the handling of personal information.

#### 6.2.9. Serverless Layer

The serverless layer implements the *store* and *query* serverless functions as shown in Figure 15 to support logging and querying operations of decisions made by the compliance verification tool using OpenFaaS [104]. OpenFaaS (or the Functions as a Service) is an open-source framework which allows building serverless functions on top of the containers.

#### 6.2.10. Scheduler

The scheduler makes use of Ofelia, a docker-based job scheduler [105], to handle time-based job scheduling tasks. Listing 1 shows the script used in our implementation to schedule the job. The time interval for the automated compliance checking can be adjusted by updating time in Listing 1. Further, in the script we can see tekactool and parser. tekactool is the docker deployment of our compliance verification tool that Ofelia scheduler depends on to run (or trigger) automated compliance check via shared service component CRON, while parser is the other docker image that the Ofelia scheduler requires.
Listing 1Ofelia scheduler docker script.ofelia:  image: mcuadros/ofelia:latest  depends_on:   - tekactool   - parser  command: daemon --docker  restart: always  volumes:   - ./:/app   - /var/run/docker.sock:/var/run/docker.sock:ro   labels:   ofelia.job-run.datecron.image: “parser”   ofelia.job-run.datecron.schedule: “@every 86400s”   ofelia.job-run.datecron.command: “python3 /app/core/cron/Cron.py”

## 7. Evaluation

This section presents the evaluation of our tool with regards to performance and scalability. We have evaluated the execution overhead of our tool’s key functionalities (consent creation, auditing, and compliance verification). Scalability testing was completed as well. This is due to the fact that our tool is a part of smashHit, and smashHit is driven by two business use cases in industries such as automobile insurance and smart cities [11]. Both use cases require a scalable solution to handle a large number of users. Our tool was also evaluated in terms of GDPR compliance to ensure that it, itself, is GDPR-compliant. Section 6.1 provides details about the testing system setup. Section 7.1 describes the performance evaluation, while Section 7.2 presents the evaluation based on GDPR.

### 7.1. Performance and Scalability

As presented in Section 7, our performance evaluation focuses on auditing, compliance verification, and consent creation. We measured the time it takes to create ten different consent instances. The required consent information was provided manually. The *X*-axis in Figure 16 represents the consent creation indicated by CS. The *Y*-axis represents the time taken in seconds. Figure 16 shows a steady curve with few rises and falls in time. The decrease in time, particularly for CS4, and the increase for CS9 and CS10, are due to the amount of payload sent while requesting to create consent. The size of the payload depends on the amount of the information present in the consent (or JSON input). For CS4, 856 bytes were sent, while for CS9 and CS10, it was around 1023 bytes, and the rest of the payload was around 950 bytes. With the increasing payload, more consent information, such as processing details, was passed. According to our observations, the average time required to create consent is approximately 7.3 s. This time period for consent creation also includes the time required for intermediate steps such as consent validation and conversion to legal KG, and also the time required for consent information encryption, as consent information is stored encrypted.

Figure 17a depicts the compliance evaluation based on consent, while Figure 17b depicts the compliance checking evaluation time based on the data provider. The data provider is a data subject who owns the data and retains the right to decide whether or not to share it with the DC (or DP) for processing. Similar to consent creation, *Y*-axis of both Figure 17a,b represents the execution time in seconds. The *X*-axis, on the other hand, represents compliance based on consent, indicated by CC in Figure 17a, and compliance based on data provider, indicated by DPC in Figure 17b. The term DPC stands for data provider compliance (or also compliance based on data provider). The observed variation in compliance checking in Figure 17a, similar to consent creation, is due to the amount of information present in the consent that must be assessed.

Overall, compliance checking (i.e., core compliance) based on consent took an average of 6.6 s. However, the time required for compliance checking varies greatly depending on the data provider, as a data provider may have multiple consents. Data providers 1 and 3 each have only one consent; thus, we observed a similar time in compliance checking as indicated by DPC1 and DPC2 in Figure 17b. Similarly, data provider 2 had two consents, so the time required for compliance checking is nearly twice (see DPC2 in Figure 17b) that of data provider 1. Other data providers such as 4, 5, and 6, exhibited a similar pattern as can be observed in Figure 17b from the graph indicated by DPC4, DPC5, and DPC6, respectively. Regardless of the number of consents, we observed a similar time to a compliance check based on consent on average, which is 6.12 s. This compliance check evaluation, however, excluded the privacy and security component as it is only called when all of the core compliance decisions are evaluated as true. As a result, we assessed privacy and security separately. In the case of granted case, the privacy and security component takes an average of 0.12392 milliseconds and 0.1234 milliseconds in a denied decision case. If security and privacy are also utilized, a compliance check takes an additional 0.12392 and 0.1234 milliseconds to that of core compliance checking time.

The evaluation for auditing based on consent and data provider, such as compliance checking, is shown in Figure 18a,b. The *Y*-axis represents time in seconds. The *X*-axis represents auditing based on consent, as indicated by FCA in Figure 18a, and auditing based on data provider, as indicated by FDPA in Figure 18b. The letter F in both figures denotes full auditing (discussed in Section 6.2.6). The full auditing is expected to take more time because it considers all information. Auditing exhibits a similar pattern to compliance checking, as shown in Figure 18a,b. Auditing based on consent took about 7 s on average, and auditing based on data provider took about 22 s. The delay in auditing based on data provider is due to checking all consents and decision logs specific to the data provider. In comparison to compliance checking based on data providers, which only checks active consent, auditing provides an audit for all consent, even if it is not active (e.g., revoked consent). Similar to the consent creation, both the compliance checking and the auditing time also involve the time for activities such as performing decryption, which is a time-consuming activity [106].

To summarize, we can observe a higher time in the performance evaluation. This is because of the associated additional time-consuming activities such as layered encryption and decryption (see Figure 3 and Figure 4). This is also a known trade-off that comes with increased security [107].

From the loosely coupled architecture of the tool and the use of scalable technologies, such as Docker and serverless, we can observe the scalability features, as this allows scaling of the individual components. Numerous studies, such as [108,109,110], have demonstrated the scalability of the loosely coupled architecture and the support provided by technologies such as Docker and serverless. Moreover, to the observed scalability of the tool from the loosely coupled architecture, we also performed a scalability testing with the Kubernetes platform, as discussed in Section 6.1. When evaluating the scalability, we only looked at resource consumption and the number of scaled pods [111] (i.e., the smallest deployable computing units that Kubernetes [73] allows you to create and manage). The horizontal pod autoscaler, which we used in our scalability testing deployment, automatically scales the number of pods into the number of replicas based on the set observed metrics such as average CPU utilization and average memory utilization [112]. Three replicas and a load balancer are used in the initial deployment. The load was generated using a load generation script. The autoscale replicas of pods were limited to 40, and the observed metrics for the scale used in our testing were average CPU utilization. The CPU utilization scalability threshold was set to 95%. We were able to successfully autoscale the application to the defined number of pods as the load generation script increased the load. The time it took to create new replicas was approximately 5–6 s. On Linode cluster 1, which also includes the load balancer, we observed nearly the maximum 324.58% CPU utilization. Similarly, on Linode clusters 2 and 3, we observed maximum CPU utilization of 96.91% and 119.16%, respectively, and I/O rates of 81.97, 28.74, and 12.60 blocks/s. However, network utilization was around 4.77 Mb/s for cluster 1 and 1.05 Mb/s and 132.36 Kb/s for clusters 2 and 3. Finally, to simulate virtual users, we used the open-source load-testing tool locust [113]. We observed a similar result for scalability with a locust, an automated load generator, simulating 52,200 users, making a maximum of 241 requests per second (minimum 100 requests per second). With the scalability testing, we observed that the tool is able to scale with the growing load (i.e., requests), proving that the application supports scalability provided that the necessary setup (i.e., configuration of appropriate scaling such as autoscaling) is made.

### 7.2. TOMs

The goal of our tool is to enable transparency in data sharing (and processing) by automatically performing tasks such as compliance verification based on informed consent granted by data subjects. In this section, we present the evaluation of the implemented TOMs (see Table 1), which is also one of our work’s contributions (see Section 1). The qualitative evaluation was conducted in accordance with the TOMs. This is also a common method of privacy evaluation used by researchers such as Ryan et al. [6] to assess their work using the Irish Data Protection Commission’s self-assessment checklist [114].

We have considered seven protection goals identified in relation to GDPR principles, as shown in Table 1. The SDMs’ protection goals are implemented in our compliance verification tool as TOMs. Table 1 contains information about the SDMs’ protection goals as well as the relevant TOMs. Both automated and manual evaluations were performed. For the automated method, the test cases were written and executed using Python’s unittest [115] test framework. The unit test was written with various conditions, with some expected to fail and others expected to pass. For example, we ran three different tests for role-based access, which is also one of the TOMs. We tested endpoint access without a valid JWT in the first test case. In the second test case, we tested endpoint access using a valid JWT token but an unauthorized role, and in the third test case, we tested endpoint access using a valid JWT token and an authorized role. We expected the test to fail in the first two cases. The failure here indicates that the application is inaccessible and returns an invalid HTTP status code as defined in the application, such as 403, indicating forbidden access, which is then asserted to be true in order to pass the unit test. When we pass everything valid in the third case, we expect the application to be accessible with the valid response and HTTP status code. To pass the test, the response is asserted to evaluate to true. Additionally, for tasks such as auditing, a manual evaluation was conducted. For example, in the case of auditing, when the audit responses were manually retrieved, a manual verification that everything was as expected was performed. Based on the testing, we were able to meet all of the SDMs protection objectives. The scope of our study’s SDM protection goals was constrained by the identified TOMs, which are listed in Table 1.

## 8. Conclusions

In this paper, we present a data protection By design tool for automated GDPR compliance verification based on semantically modeled informed consent. We also present a novel systematic approach for translating the GDPR legislation into code while considering industrial requirements such as interoperability and scalability (see Table A1). This includes the systematization of the steps that are required for automating compliance checks and implementing necessary security measures (i.e., such as TOMs). Further, we present the technical factors that must be considered in order to satisfy industrial requirements. Our tool follows a microservices architecture, which supports scalability and flexibility and utilizes semantic technology (e.g., KG) to facilitate data interoperability. By conducting scalability testing (i.e., testing with simulated users and checking if the tool scales with the growing number of users) (see Section 7.1), we have validated our tool’s scalability. The performance evaluation (see Section 7.1) provides information on the associated overhead for tasks such as compliance checking and consent creation, which is useful for the future improvements of the tool. The conducted GDPR TOMs evaluation (see Section 7.2) has confirmed that GDPR requirements, such as confidentiality and the adoption of data protection by design principles, have been satisfied and utilized. Finally, the comparative analysis with existing work, presented in Table A1, indicates that our work advances the field by addressing the limitations of current studies (see Section 3) and industrial needs. As a result of the collaboration with legal experts and industrial partners, our work can support SMEs, which usually lack the necessary resources to implement their own GDPR compliance tools. The source code of our tool is publicly available on GitHub [116].

While our work can be generalized to other domains, it was driven by specific use cases (see Section A.2) and thus may fall short of requirements from other domains. We consider this as one of its limitations. Another limitation is the compliance performance bottleneck in the long run due to the growing number of compliance checks that must be performed.

In addition to addressing the existing challenges (see Section 3), we have also learned some valuable lessons. These lessons include the following:Translating regulations into a machine-readable format requires a considerable effort with regard to the collaboration between legal and technology experts. Both the law and computer science fields necessitate the precise definition of concepts. The translation of these concepts to a machine-readable format needs to be precise as well.A well-designed core semantic model is key to achieving a common understanding of specific information across different systems. However, the integration of existing ontologies is challenging, as each ontology presents concepts from a different perspective.The early and consistent collaboration with business (or industry) use case partners and legal experts helps understand the requirements and implications from both perspectives. This helps the system design and implementation adapt to business needs, while fulfilling legal requirements.The use of semantic technology has shown to have various benefits for our work. However, existing and well-established industry systems do not always utilize semantics. This required us to to adapt our implementation to the industry requirements. To ease the process, we have selected a JSON format to communicate the consent information and compliance check results.Scalability, interoperability, and the simplification of the integration processes are critical for use case partners, as is the tool’s consent creation functionality. Any tool’s design and implementation must be tailored to such requirements.

In conclusion, the main benefits of our work are (i) systematization of the translation of legal requirements into code following specific industrial needs (i.e., such as the problem of broken consent chains), (ii) generalizability into other data sharing domains and regulatory frameworks, and (iii) lessons learned, which would benefit anyone working in this domain, particularly new researchers and developers. We have also addressed existing limitations such as interoperability and scalability. Future work will include the following: (i) extending the work to other data-sharing domains and regulatory frameworks; (ii) improving performance of the tool by parallelizing core processes. A possible direction in relation to our future work could be to incorporate additional functionalities, such as audit search, and to incorporate other GDPR legal bases, such as contracts.

## Figures and Tables

**Figure 1 sensors-22-02763-f001:**
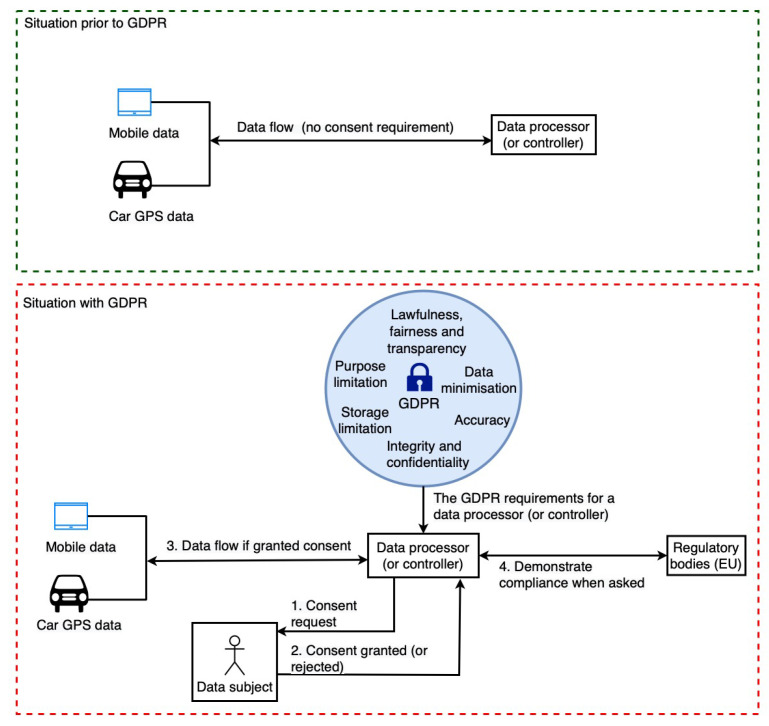
The areas of concern that need to be focused on in light of the changing data sharing/processing landscape as a result of GDPR implementation.

**Figure 2 sensors-22-02763-f002:**
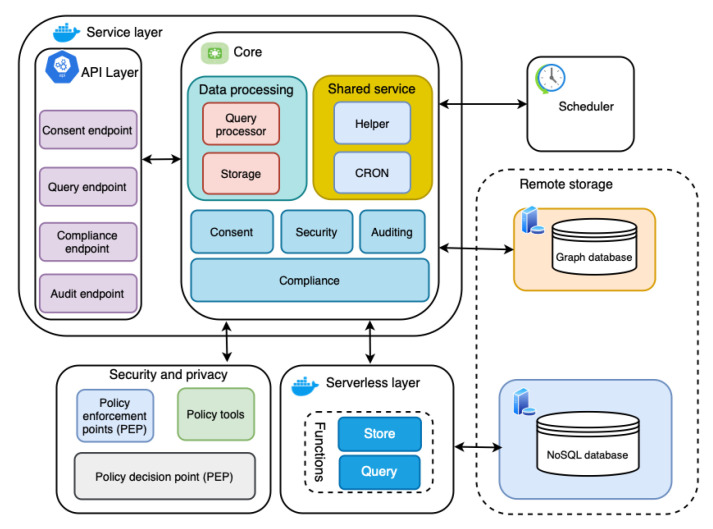
Architecture of the automated GDPR compliance verification tool.

**Figure 3 sensors-22-02763-f003:**
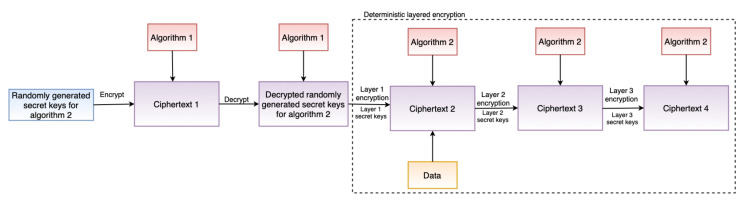
Architecture of encryption scheme.

**Figure 4 sensors-22-02763-f004:**
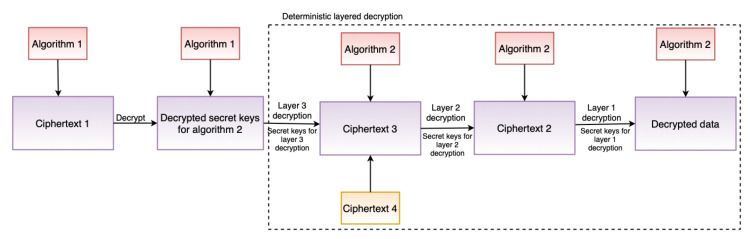
Architecture of decryption scheme.

**Figure 5 sensors-22-02763-f005:**
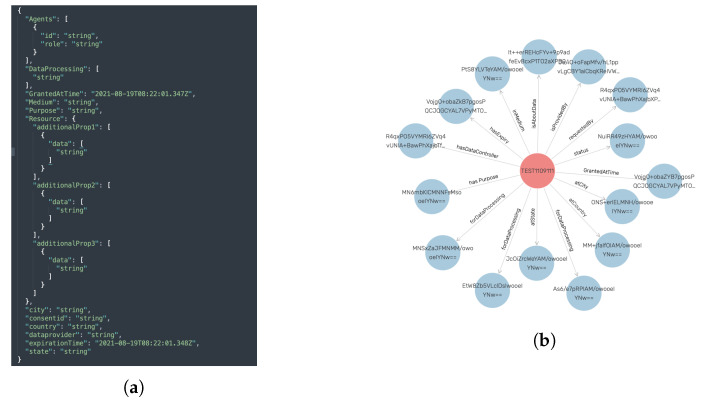
A snapshot of the JSON schema used for consent and the consent module’s creation (or representation) of consent in the legal KG. (**a**) Consent JSON schema. (**b**) KG representation of consent in GraphDB after successful validation.

**Figure 6 sensors-22-02763-f006:**
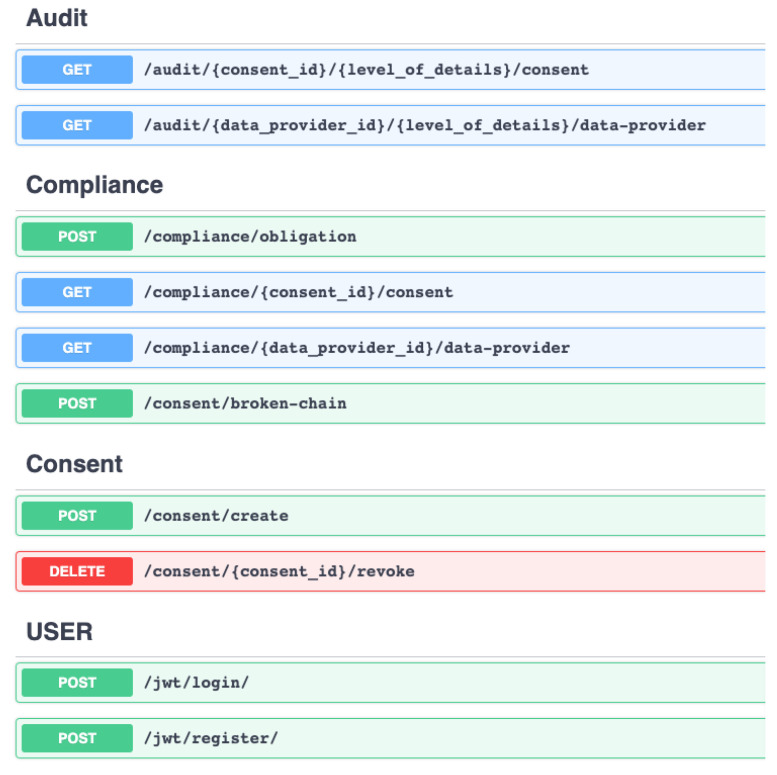
A snapshot of the REST API’s endpoints in Swagger.

**Figure 7 sensors-22-02763-f007:**
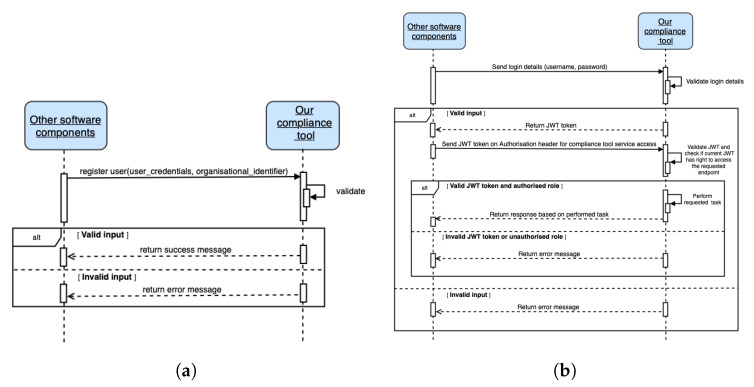
Sequence diagram for JWT user registration and role-based endpoint access by other software components. Any third-party application that interacts with our tool is represented by the other software components. (**a**) Sequence diagram JWT user registration. (**b**) Sequence diagram for role-based service access.

**Figure 8 sensors-22-02763-f008:**
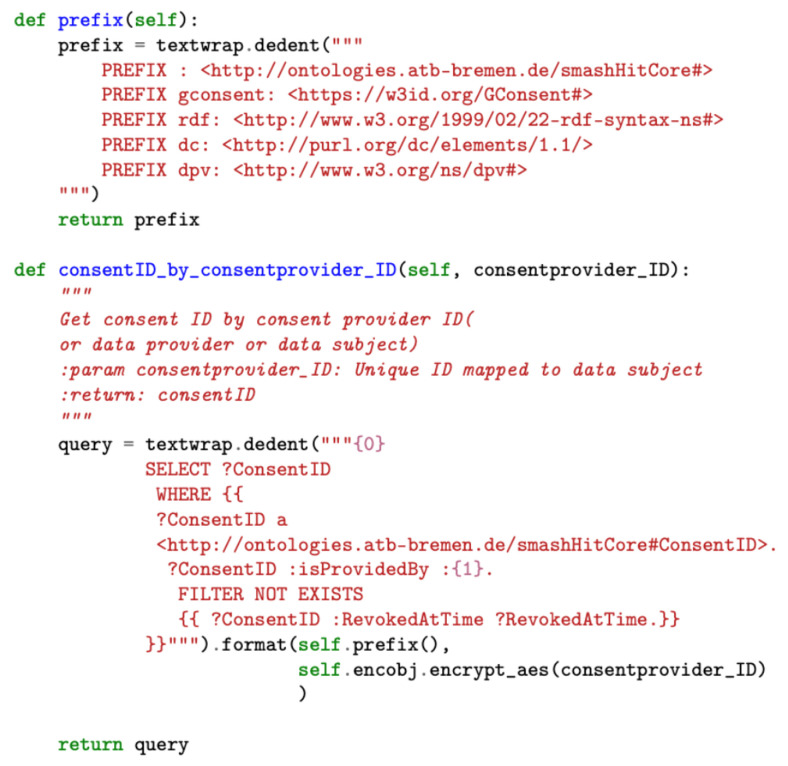
A snippet of code from the query processor module.

**Figure 9 sensors-22-02763-f009:**
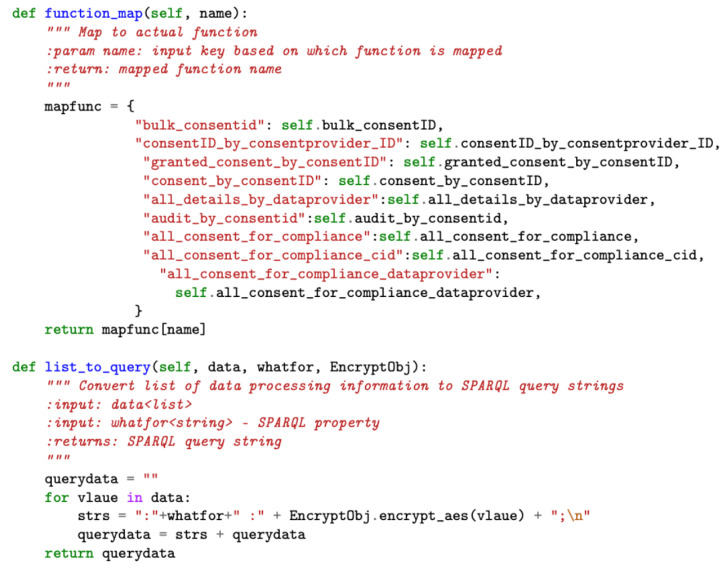
Sample code snapshot of the helper component of shared service module.

**Figure 10 sensors-22-02763-f010:**
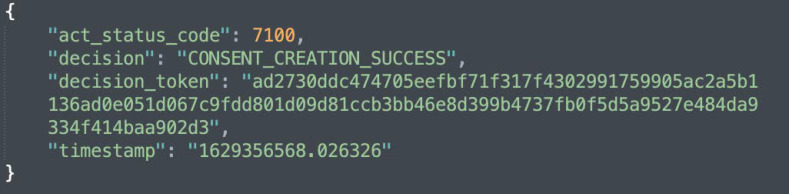
A snapshot of the consent creation response.

**Figure 11 sensors-22-02763-f011:**
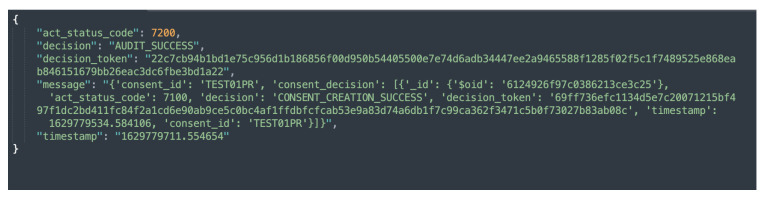
Sample response to a request for partial auditing based on consent.

**Figure 12 sensors-22-02763-f012:**
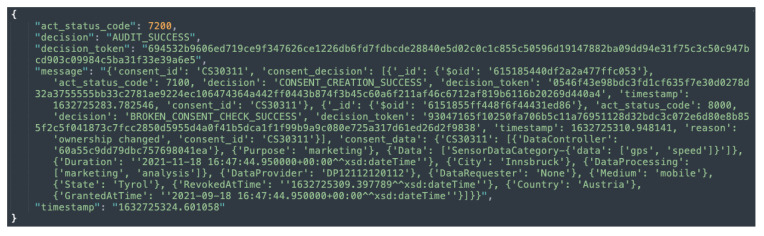
Sample response to a request for full auditing based on consent.

**Figure 13 sensors-22-02763-f013:**
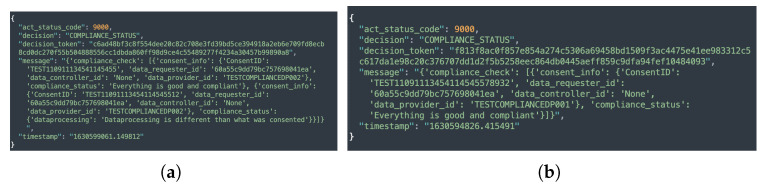
Compliance check response. (**a**) Compliance check based on data provider. (**b**) Compliance check based on a single consent.

**Figure 14 sensors-22-02763-f014:**
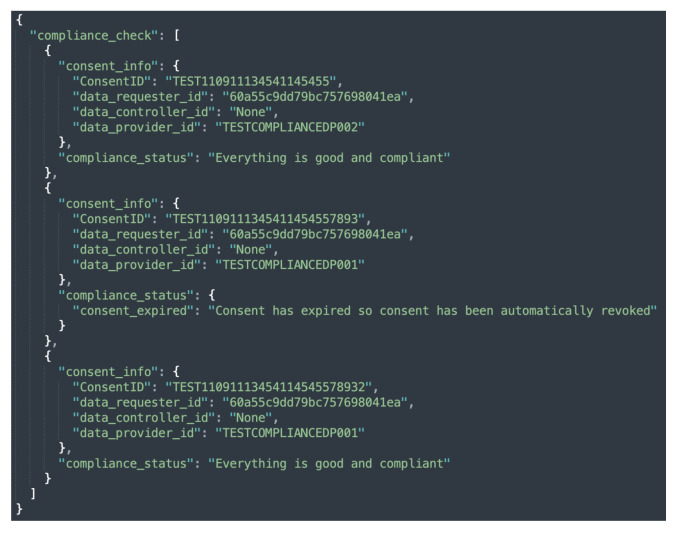
Sample response for an automated compliance check.

**Figure 15 sensors-22-02763-f015:**
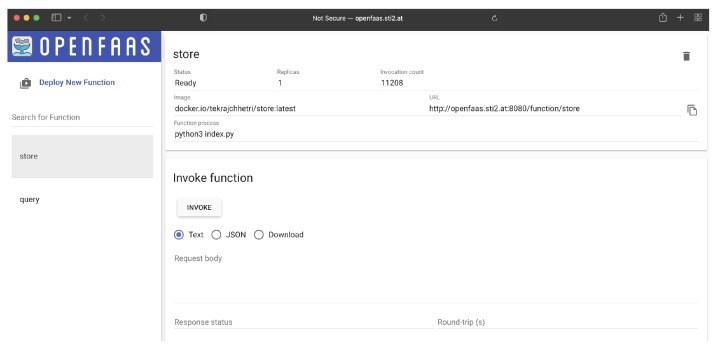
Deployed OpenFaas functions.

**Figure 16 sensors-22-02763-f016:**
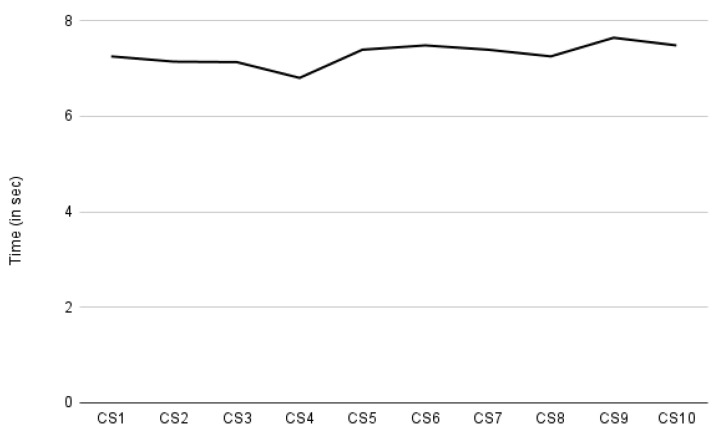
Time spent on consent creation.

**Figure 17 sensors-22-02763-f017:**
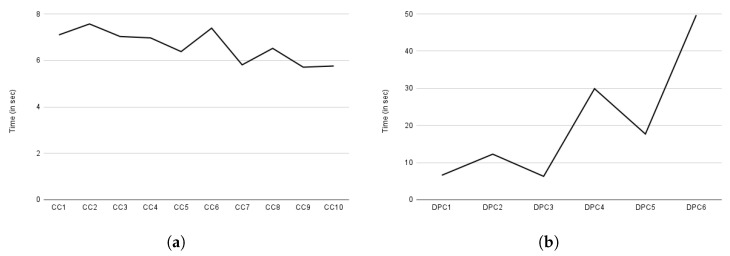
Time spent on compliance checks. (**a**) Execution time for compliance check based on consent. (**b**) Execution time for compliance check based on data provider.

**Figure 18 sensors-22-02763-f018:**
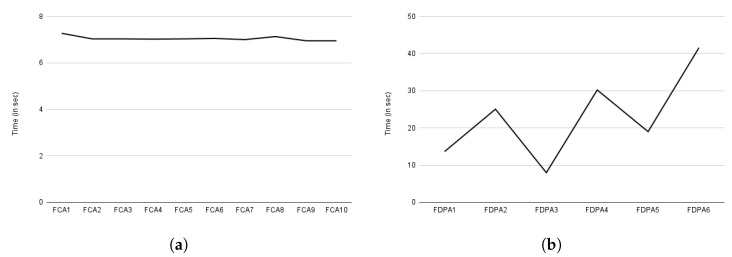
Time spent on auditing. (**a**) Execution time for full audit based on consent. (**b**) Execution time for full audit based on data provider.

**Table 1 sensors-22-02763-t001:** GDPR principles and associated SDM protection objectives, their scope and purpose, as well as the associated technical–organizational measures (TOMs) and their implementations in our compliance verification tool.

GDPR Principles	SDM Protection Goal	Scope and Purpose	TOM	Our TOM Implementation	Our Tool’s Component
Purpose limitation (Art. 5(1)(b))	Unlinkability	Takes the principle of purpose limitation into account; defines the permissible purpose changes.	Role concepts with graduated access rights on the basis of identity management and secure authentication process.	JavaScript Object Notation (JSON) [18] Web Tokens (JWT) [19]-based access control is implemented. Further, customization of standard JWT token-based access is implemented, enabling role-based endpoint access.	API layer
Storage limitation (Art. 5(1)(e))	Availability	Ensuring the availability of data at a certain time for those who require it at that time.	Documentation of data syntax.	PEP-8 [20] coding convention is followed throughout the entire code base; code is commented for better understandability. Swagger (an interface description language for describing RESTful APIs) is used to document the RESTful APIs.	All components
Lawfulness, fairness and transparency (Art. 5(1)(a))	Transparency	The extent and the form in which data processing should be kept transparent towards data subjects and supervisory authorities; information and disclosure obligations pursuant to Art. 12 et seq. GDPR, the notification obligation pursuant to Art. 34 GDPR, the documentation of the processing pursuant to Art. 30 GDPR.	Documentation of consents, their revocations and objections.	Consent and their states, such as revocations, are stored in the GraphDB [21] database, as well as logged in the MongoDB [22] database.	Compliance, Consent, Audit
Accuracy (5(1)(d)), Integrity and confidentiality (Art. 5(1)(f))	Intervenability	The extent to which data subject rights are to be granted; how data subjects can exercise their rights, how to ensure that requests are made in a legitimate manner, what corrections can be taken in the processing of personal data (e.g., by rectification, erasure, or limitation of the processing of personal data) and in what form data can be transferred by or to other controllers.	Operational possibility of compiling, consistently rectifying, blocking and erasure of all stored personal data.	All data are stored in the KG with a unique ID and these data are processed via REST API endpoints; by providing personal information that resolves to a unique ID, users can access their data via a user interface.	Compliance, Consent
Integrity and confidentiality (Art. 5(1)(f))	Confidentiality	Takes care that the disclosure of certain data is denied to those who are not authorized to have access to it; takes into account the processes, systems, and services potentially vulnerable to unauthorized access.	Encryption of data.	Deterministic searchable encryption technique is used to encrypt the data. The Rivest–Shamir–Adleman (RSA) [23] with Public Key Cryptography (PKCS) Standards) #1 Optimal Asymmetric Encryption Padding (OAEP) is used for asymmetric Advanced Encryption Standard (AES) [24] session key encryption and chained three layers AES for data encryption. Further, implementation of authentication procedure. & Identity management is used to ensure that only registered components have access to endpoints.	Security
Integrity and confidentiality (Art. 5(1)(f)), Accountability (Art. 5(2))	Integrity	Ensuring that data related to an identified or identifiable person are kept intact and up-to-date; ensuring that the processes, systems, and services are correctly planned, operated, and controlled according to the intended purpose.	Protection against external influences.	Security measures such as encryption, role-based access controls to prevent unauthorized data access. Audits and tests to document functionality, risks, security gaps.	Security, Audit
Data minimization (Art. 5(1)(c))	Data minimization	Implementation of the data minimization requirement of the GDPR; establishment of retention periods for personal data and processes to ensure compliance.	Reduction of non-required attributes of data subjects.	Consent creation REST API endpoint defines minimal set of variables for processing.	Consent

## Data Availability

Not applicable.

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
