# Peer review of "Data Protection by Design Tool for Automated GDPR Compliance Verification Based on Semantically Modeled Informed Consent"

_sensors, 2022, doi:10.3390/s22072763_

Round 1

Reviewer 1 Report

First of all, I have to state that your paper is very long with redundant and unnecessary content, which makes it not easy to read and understand for scientists outside of the field of research. Although, this paper has proposed a solution to a challenging issue related to automated GDPR compliance based on semantically modelled informed consent, I have mixed feelings about this paper. The authors have shown the implementation and demonstrated the effectiveness of the proposed tool in the insurance and smart cities use cases, but the overall delivery is confusing. The following state the reasons of my perplexity:

  1. Major:
  • The overall manuscript (in its current version) is poorly written and needs to be reorganized and improved scientifically in all sections. After reading the paper, it is still not clear about the main objective(s) of this research and I must admit that I couldn’t really appreciate the contribution(s) claimed in this paper. The authors should clarify about the main object(s) and contribution(s) of this study in a pertinent way. The research contribution(s) should be emphasized throughout the paper and mainly in the introduction and conclusion.

  • The introduction is very long, the authors are advised to remove redundant and unnecessary content to make it brief and consistent, as a summary of the paper.

The introduction should briefly place the study in a broad context and highlight why it is important. It should define the purpose of the work and its significance. The current state of the research field should be carefully reviewed and key publications cited. Please highlight controversial and diverging hypotheses when necessary. Finally, briefly mention the main aim of the work and highlight the principal conclusions.

  • For easy understanding of the importance and utility of this study, please elaborate concisely in a separate section key concepts and theoretical background of the research, or discuss them in more consistent manner while reviewing the related work.

  • The literature review of existing works is poorly presented. The authors are advised to revise this section in a more concise and effective manner. To make extensive the literature review section, please have a look at the following papers for automated GDPR compliance verification and compare them with the approach used for your proposed tool. For consistency, it could be good to combined Table 1 and Table 6 into a single table that summarizes the state-of-art compared with the present work.

  1. Tokas, Shukun, Olaf Owe, and Toktam Ramezanifarkhani. "Static checking of GDPR-related privacy compliance for object-oriented distributed systems." Journal of Logical and Algebraic Methods in Programming 125 (2022): 100733.
  2. Hamdani, Rajaa El, et al. "A combined rule-based and machine learning approach for automated GDPR compliance checking." Proceedings of the Eighteenth International Conference on Artificial Intelligence and Law. 2021.
  3. Barati, Masoud, and Omer Rana. "Tracking GDPR compliance in cloud-based service delivery." IEEE Transactions on Services Computing (2020).
  4. Mahindrakar, Abhishek, and Karuna Pande Joshi. "Automating GDPR Compliance using Policy Integrated Blockchain." 2020 IEEE 6th Intl Conference on Big Data Security on Cloud (BigDataSecurity), IEEE Intl Conference on High Performance and Smart Computing, (HPSC) and IEEE Intl Conference on Intelligent Data and Security (IDS). IEEE, 2020.
  5. Bonatti, Piero A., et al. "Machine Understandable Policies and GDPR Compliance Checking." KI-Künstliche Intelligenz3 (2020): 303-315.
  6. Truong, Nguyen Binh, et al. "Gdpr-compliant personal data management: A blockchain-based solution." IEEE Transactions on Information Forensics and Security 15 (2019): 1746-1761.
  7. Ranise, Silvio, and Hari Siswantoro. "Automated legal compliance checking by security policy analysis." International Conference on Computer Safety, Reliability, and Security. Springer, Cham, 2017.
  8. Robol, Marco, Mattia Salnitri, and Paolo Giorgini. "Toward GDPR-compliant socio-technical systems: modeling language and reasoning framework." IFIP Working Conference on The Practice of Enterprise Modeling. Springer, Cham, 2017.

System design and methodology

  • The present work and model seem engineering based research, there is no evidence of mathematical models, study metrics and constraints. The absence of any formalization of the proposed system model and methods in a formal or mathematical notations make the manuscript to look as it is written in a technical report format, but not in a scientific fashion.

  • The authors are advised to clarify who are the users of the proposed tool referring to the GDPR? And in which situation it can be used? Considering the statement on Page 3 line 90-91, is the proposed tool designed to be used only in smart cities and insurance use cases or it can also be generalized to support other use cases? The authors are advised to briefly elaborate on this point.

  • Page 8, Subsection 3.2 – It is not clear how the consent is captured from the data subject(s) using the KG approach with the smashHitCore ontology? What is the difference between the consent forms and the compliance forms?

  • Based on the statement in Section 4, page 9 line 302-303. Could you elaborate on why the micro service architecture has been adopted for the proposed tool? And how the micro service architecture pattern can resolve the scalability problem, and address the industrial requirements?

  • Page 10 – Referring to the Table 4, what is the purpose and role of the TOMs and why they are important in this research and how reliable is their legal perspective evaluation provided? The authors are advised to elaborate on these points in the manuscript. As they claimed later in the Section 5.3 (page 31 line 854-855) that they have presented the evaluation from a legal perspective, which is one the contribution of this study.

  • Page 13 – In Figure 4, the authors should be specific about the software components that interact with the proposed tool instead of just mentioning other software components, which is really vague. In addition, the Figure 4 caption should provide a brief description of the figure 4a and 4b.

  • Page 17 – It is not clear in Algorithm 1, what is considered to decide whether the consent is valid or not based on a given JSON consent model?

  • Page 17 line 439-440 – What is the reason for you to combine asymmetric and symmetric encryption techniques for security module implementation? Please, clarify this point.

  • Page 22 – Please revise Algorithm 3 and 4 (as your main contributions) to present them in a more effective manner and highlight how the compliance verification and checking operations are systematically and automatically handled.

Implementation, experiment and evaluation

  • The implementation, experiment and evaluation sections should provide a concise and precise description of the experimental results, their interpretation, as well as the experimental conclusions that can be drawn. What can we practically do with the experiment’s results? How the proposed work performs better than existing solutions?

  • Considering the statement on Page19 line 489-490, could the authors elaborate on how data controllers or processors can use auditing information to systematically demonstrate compliance? How the proposed approach can detect a consent violation to trigger a notification? As mentioned on Page 19 line 511-513.

  • Considering the compliance model provided in equation (1) page 20, could you break it down and detail your compliance algorithm mathematical or logic model?

  • It is not clear how the accountability and auditability requirements are addressed in this research? Moreover, the authors failed to demonstrate how the scalability issue has been addressed in this study. Based on the claims mentioned in the conclusion section (Page 33, line 934-938). Is the fact of using Docker and serverless approach that makes this tool scalable? Which metrics did you used and how did you measure the scalability of your proposed tool?

  • The authors should elaborate about which metrics and methods you used to evaluate the satisfaction of SDM protection goals claimed in table 5, page 32.

  • Page 33, Table 6 – Please extend the comparison of your proposed solution with all relevant existing studies, such as [8], [35], and other newly suggested references. Check also the consistency of the all content in this table. See more comments in the reviewed pdf file.

  • The finding(s) and limitation(s) of the study are ignored in this manuscript. What are the main findings or lessons learned and what are remaining challenges or limitations of this study? What has been shown that can be useful for theoretical background in future researches? Please also supplement the conclusions with enough suggestion or orientation for further research.
  1. Minor:
  • References should be numbered in order of appearance in the text. Figures and Tables should be placed in the main text near to the place where they are cited for the first time. In addition, abbreviations and symbols should be described the first time they are used in the text.

  • Page 3 line 92. The sentence is not clear; please elaborate more on this contribution.

  • Page 3 line 105. “Section 3 provides information about the methods.” It is not clear which methods the authors are referring to? Please rephrase this sentence to make it clear.

  • Page 3 line 118-121. These sentences are not clear; please rephrase them.

  • Page 3 line 124-128. It is not clear what you would like to mention there.

  • Duplicated references (e.g., the same reference is mentioned as [8] and [34]).

  • Page 9 – It could be better to combine table 3 and table 4 in a single table and revise the consistency of the table’s content.

  • Page 12 – In Figure 2, it could be better to provide an example of an API response message with realistic values for each attribute instead of putting their explanation which should elaborated in the main text.

  • Please see additional comments, questions and suggestions in the reviewed manuscript pdf file.

I suggest the authors to work on the delivery of the research, reorganize the manuscript and reduce the overall length, and clarify the purpose of their work, its importance and significance, as well as the usefulness of their research. Finally, the limitations and proposal for future research direction have to be extensively discussed. Moreover, an extensive editing and review of English language and style is required.

Author Response

Dear Reviewer,

We would like to express our gratitude for taking the time to review our article. We have updated the manuscript, taking your feedback into account.  All the modifications (or updates) are highlighted in red (in the manuscript). The major modifications made in light of your feedback are highlighted below.

  1. Restructuring of the paper to separate the implementation details from the architectural approach.
  2. Addition of the new background information  (i.e., new section) highlighting the need and importance of our work.
  3. Revision of the conclusion, to add lessons learned and limitations, which highlights the benefits of our work. 
  4. Restructuring of the introduction to make it shorter and concise and title (i.e., update based on the on the other reviewer’s comment).
  5. Addition of the review of the new state-of-the-art works.

Furthermore, the attached file contains the update's details and the response to the raised question.

Regards,

Tek Raj Chhetri

(On behalf of all the authors)

Reviewer 2 Report

This paper addresses an important and timely topic, as many organizations need solutions to deal with the compliance of their personal data processing operations with data protection regulations as GDPR. The authors report the results of a big EU-funded project (smashHit) that has researched on the topic, providing relevant contributions that can be applied to this end.

However, while the authors describe some of these valuable contributions e.g. for automated consent compliance checking, there are so much information and different topics mixed up in this paper that it makes difficult to grasp and appreciate these actual and relevant ones.

This is highlighted when, for example, checking the description of the related work. First, there is a state of the art section which reports many different, sometimes disconnected previous works, from very general approaches to specific tools. Some works propose just some guidelines, or a set of requirements, and as such I think a proper comparison with a tool cannot be carried out. Then, in section 5.4 they go back to some of these results (once they have presented their tool) to compare them again with their solution, which is quite confusing. Why not having all in a single place?

Furthermore, I would suggest authors to review the terminology they use. They mix up privacy and data protection concepts. For example, they refer to 'privacy by design', which is an approach for the development of privacy-friendly systems, but link that to GDPR principles, when the GDPR actually does not mention 'privacy by design', but data protection by design.

The paper is sometimes hard to read and follow. I would recommend having it checked so as to improve readability.

Author Response

Dear Reviewer,

We would like to express our gratitude for taking the time to review our article. We have updated the manuscript, taking your feedback into account. The attached file contains the update's details and the response to the raised question.

Regards,

Tek Raj Chhetri

(On behalf of all the authors)

Round 2

Reviewer 1 Report

The authors have addressed most of my previous comments and revised the manuscript accordingly. It can be seen that they have tried their best to improve the manuscript. However, there are still issues that need to be addressed for the paper to be suitable for publication. Please see the review below.

-----------

The authors have addressed most of my previous comments and revised the manuscript accordingly. It can be seen that they have tried to their best to improve the manuscript. However, the following issues still need to be addressed for the paper to be suitable for publication:

  1. The authors are advised to improve the overall writing skills and review the English language and style of this manuscript for suitable presentation quality and scientific soundness. The paper needs thorough proofreading and most of the sentences need to be rephrased consistently.

  1. The paper length is still too long. It could be better to make it more compacted by removing useless and redundant contents that make it difficult for reading.

  1. For easy understanding of the readers, could you elaborate more on the meaning you are giving to the concept of "scalable" in your contribution claims? (Lines 87-88)

  1. Considering the statement on Lines 92-93, could you briefly elaborate here on the evaluation metrics considered, as well as the results (higher level overview)?

  1. Considering the claim on Lines 96-97, could you elaborate on how the consent revocation is systematically handled and how the broken consent agreement chains can be reconstructed with the proposed tool? Spelling issue “too” --> “tool” in the sentence on Lines 96-97.

  1. The Related Work section is still too long with many useless details. Could you make it more brief and consistent? As you have already summarized in Table 1 reviewed state-of-the-art projects

  1. None of the smart city or insurance use case scenarios have been properly described anywhere in the manuscript. Please elaborate effectively on that to show how the proposed solution will be used in these use cases as claimed in your contributions.

  1. The figure 1 name sentence is not clear (Page 4). It needs to be reformulated or rephrased. Could revise effectively Figure 1 by clarifying the situation prior to GDPR and the situation with GDPR. A good idea would be to design this figure considering the two use cases (smart city and insurance) mentioned in the introduction.

  1. Line 330 - The section name needs to be labeled more explicitly to give higher-level understanding of its content. E.g. “Methodology”, "Research methods" or “Methods for ...”, etc.

  1. You have provided only Algorithm 1 (Role-based API endpoint access) and Algorithm 2 (Consent validation for consent creation), I cannot find any consent agreement by the data subject and GDPR compliance algorithms in the paper. Could you provide the algorithms of the proposed consent agreement by the data subject and automated GDPR compliance verification? Especially, the automated GDPR compliance verification algorithm must be provided as it is claimed to be one of the main contributions of this paper.

  1. Referring to line 360, the mentioned “v1 Table 3 and v2 Table C2” tables cannot be found in the manuscript, as well as in the referenced paper [57]. Could you fix that?

  1. Considering this statement in Line 382-384, how is the consent evaluated against GDPR requirements? Is this verification process manual or automatic process supported by the system?

  1. Could you elaborate more on the consent validation mechanism in Algorithm 2 (Subsection 5.1.5., Lines 454-460)? What are you taking into consideration to validate a consent json object or instance? Could you break down the validate_json_consent(), transform_and_map_to_legal_kg(json_consent) and create_consent(map_consent)  functions in your proposed Algorithm 2? Because readers cannot learn anything from the current version of Algorithm 2.

  1. Considering the compliant behavior examples given in Subsection 5.1.7 (Lines 513-542). For easy understanding, it could be better to use a figure to illustrate these examples with a brief explanation.

  1. Considering Subsection 6.1, for simplicity and readability, it could be better to summarize the experimental setup parameters and details in a table that should be briefly explained.

  1. Considering the following references, they are well-known and widely used commercial production websites. I think, it is not required to mentioned them exaggeratedly in the reference list. It could be appropriate to reference the related scientific publications or technical reports/docs if needed.

* reference numbers in you paper:  32, 41, 79, 97 ~ 102

Author Response

Dear Reviewer,

We would like to express our gratitude for taking the time to review our article. We have updated the manuscript, taking your feedback into account.  All the modifications (or updates) are highlighted in red (in manuscript).  Furthermore, the update's details and the response to the raised question, which are highlighted in italics, are included in the attached file.

Regards,

Tek Raj Chhetri(On behalf of all the authors)
